# Identification of Six Phytochemical Compounds from *Asparagus officinalis* L. Root Cultivars from New Zealand and China Using UAE-SPE-UPLC-MS/MS: Effects of Extracts on H_2_O_2_-Induced Oxidative Stress

**DOI:** 10.3390/nu11010107

**Published:** 2019-01-07

**Authors:** Hongxia Zhang, John Birch, Jinjin Pei, Isam A. Mohamed Ahmed, Haiyan Yang, George Dias, A. M. Abd El-Aty, Alaa El-Din Bekhit

**Affiliations:** 1State Key Laboratory of Biobased Material and Green Papermaking, College of Food Science and Engineering, Qilu University of Technology, Shandong Academy of Science, Jinan 250353, China; 2Department of Food Science, University of Otago, P.O. Box 56, Dunedin 9054, New Zealand; john.birch@otago.ac.nz; 3Shaanxi Key Laboratory of Bioresources, Shaanxi University of Technology, Hanzhong 723000, China; xnjinjinpei@163.com; 4Department of Food Science and Nutrition, College of Food and Agricultural Sciences, King Saud University, Riyadh 4545, Saudi Arabia; iali@ksu.edu.sa; 5College of Food and Pharmacy Sciences, Xinjiang Agricultural University, Urumqi 830000, China; yanghaiyan@163.com; 6Department of Anatomy, University of Otago, P.O. Box 56, Dunedin 9054, New Zealand; george.dias@otago.ac.nz; 7Department of Pharmacology, Faculty of Veterinary Medicine, Cairo University, Giza 12211, Egypt; abdelaty44@hotmail.com; 8Department of Medical Pharmacology, Medical Faculty, Ataturk University, 25240 Erzurum, Turkey

**Keywords:** *Asparagus officinalis* L. roots, bioactive compounds, ultrasound assisted extraction, solid-phase-extraction, ultra-performance liquid chromatography-tandem mass spectrometry, cell culture

## Abstract

A simple, rapid, specific, and sensitive method was developed for the simultaneous identification and quantification of six major bioactive compounds, namely, caffeic acid, quercetin, apigenin, ferulic acid, baicalein, and kaempferol, from *Asparagus officinalis* roots (ARs) native to New Zealand (green and purple cultivars) and China (yellow, green, purple, and white cultivars) using ultrasound-assisted, solid-phase extraction (UASE-SPE) coupled with ultra-performance liquid chromatography-tandem mass spectrometry (UPLC-MS/MS). The method was validated in terms of linearity, limit of detection (LOD), limit of quantification (LOQ), accuracy (expressed as recovery %), and precision (expressed as relative standard deviation (%RSD)). The retention times, ultraviolet visible (UV-vis) data, and mass spectral patterns of the detected peaks matched those of commercial standards, allowing characterization of the target compounds. The LODs and LOQs were 23 ng/mL and 70 ng/mL, 50 ng/mL and 150 ng/mL, 10 ng/mL and 30 ng/mL, 18 ng/mL and 54 ng/mL, 14.4 ng/mL and 43.6 ng/mL, and 7.5 ng/mL and 22.5 ng/mL for caffeic acid, quercetin, apigenin, ferulic acid, baicalein, and kaempferol, respectively, and the mean recovery rates were 85.8%, 73.0%, 90.2%, 80.6%, 76.7%, and 74.5% for the six compounds, respectively. The levels of the target compounds were significantly different (*p* < 0.05) among the six cultivars. The Chinese yellow AR had the highest levels of bioactive compounds: 6.0, 3.9, 0.4, 1.0, 0.86, and 0.8 mg/g for caffeic acid, quercetin, apigenin, ferulic acid, baicalein, and kaempferol, respectively. The AR extracts showed protective effects against oxidative stress in the HepG2 and L929 cell lines. The results indicate that AR extracts contain high flavonoid levels that provide protective functions against oxidative stress and support the potential commercial application of AR extracts.

## 1. Introduction

*Asparagus officinalis* L. (green asparagus) is consumed worldwide as a popular fresh vegetable due to its nutritional value and high levels of bioactive phenolic compounds [1]. Some bioactive compounds, such as inosine, rutin, and quercetin, have been found in *Asparagus officinalis* root (AR) after extensive and laborious phytochemical analysis using liquid-chromatography-mass spectrometry (LC-MS) and nuclear magnetic resonance (NMR)-based techniques [2]. Various techniques, including thin-layer chromatography (TLC) [3,4], finger-printing chromatography [1], gas chromatography-mass spectrometry (GC-MS) [5], column chromatography (CC), NMR spectroscopy, and reversed-phase high-performance liquid chromatography (RP-HPLC) [2,6], have been used for the separation and purification of asparagus root compounds, mostly from *A. racemosus*, and very few studies have investigated AR [2,4]. RP-HPLC is the most commonly used technique for analysis of flavonoids from plants because of the low volatility of these compounds [7]. Further, the use of ultrasound and microwave assisted extraction techniques to maximize the bioactivity or the yield have been reported [8]. Huang et al. [2] detected caffeic acid, ferulic acid, inosine, rutin, quercetin, and several polysaccharides with 70% ethanol extracts of AR after successive partitioning with petroleum ether, ethyl acetate (EtOAc), and *n*-butanol (*n*-BuOH). The EtOAc extract was loaded onto a silica gel column and eluted with a gradient of CHCl_3_–MeOH to afford 14 fractions. Further separation using silica gel CC (silica gel chromatographic column), Sephadex LH-20 (Hydroxypropyl glucan gel), ODS (octadecylsilyl), CC (chromatographic column), and preparative HPLC to separate different compounds that were identified using NMR spectroscopy. This conventional procedure for the identification of compounds is complicated and laborious. To date, comprehensive identification of bioactive compounds in AR has not been reported. HPLC-diode array detectors combined with mass spectrometry (MS) can afford comprehensive qualitative and quantitative data and provide a simple approach for compound identification using spectral characteristics and atom-probe tomography (APT).

Liver cancer is the fifth most frequent cancer in the world [9], and L929 has been recommended by standard institutions [10] to be used as a reference cell line for cytotoxicity evaluation of plant extracts. Thabrew et al. [11] reported a good correlation between the concentration of *Osbeckia aspera* leaf extract and protective ability against damage to HepG2 cells induced by bromobenzene and 2,6-diMeNAFQI. To the best of our knowledge, the bio-protective capacity of asparagus root extracts has been reported for *A. racemosus*, *A. cochinchines*, *A. pubescens*, and *A. africanus* [12], and there are no studies available on the bioactive compound profiles of green and purple cultivars of AR from New Zealand and purple, white, green, and yellow cultivars of AR from China. The overarching objective of this study is to provide useful information regarding the composition of bioactive compounds in AR cultivars available in New Zealand and China and to establish a strong foundation for the use of AR extracts in functional foods and biotechnological applications. To achieve this objective, we developed and validated a rapid, simple, specific, and sensitive method to quantify six bioactive compounds (caffeic acid, quercetin, apigenin, ferulic acid, baicalein, and kaempferol) in AR using ultrasound-assisted solid-phase extraction coupled to ultra-performance liquid chromatography-tandem mass spectrometry (UAE-SPE-UPLC-MS/MS). Furthermore, the efficacy of the extracts against oxidative stress was examined using an in vitro system.

## 2. Materials and Methods

### 2.1. Instruments

Solid-phase extraction (SPE) is an efficient method for sample preparation, and online SPE-LC-MS/MS was used for separation and purification of the bioactive compounds that were extracted via ultrasound-assisted extraction (UAE) under optimal conditions. Moreover, the Hep G2 and L929 cell lines were used to determine the bio-protective ability of AR extracts against hydrogen peroxide (H_2_O_2_)-induced oxidative stress.

### 2.2. Chemicals and Reagents

Ultrapure water was prepared using a Millipore Direct-Q^®^ 3 system (Millipore Corp., Burlington, MA, USA). Caffeic acid (98%), quercetin (98%), apigenin (97%), ferulic acid (99%), baicalein (98%), and kaempferol (97%) were supplied by Tianjin Biotechnology Co., Ltd. (Tianjin, China). Formic acid, ethanol, and methanol were obtained from Fisher (St. Louis, MO, USA). Hepatocellular carcinoma (HepG2) and mouse fibroblast (L929) cell lines were obtained from stocks stored at the Anatomy Department of University of Otago (Dunedin, New Zealand). Silica-based C-18 SPE cartridges, an Agilent Strata-C18 Bond Elut cartridge, Oasis HLB columns, and Supra-Clean (SI-S) cartridges (500 mg, 6 mL) were supplied by Varian (Walnut Creek, CA, USA), Agilent (Milford, MA, USA), and Biocomma (Guangzhou, China). All other chemicals and reagents were of HPLC grade.

### 2.3. Calibration Standard Preparation

Stock solutions of caffeic acid, quercetin, apigenin, ferulic acid, baicalein, and kaempferol at 1000 ng/mL were prepared in absolute methanol. The solutions were stored at 4 °C in amber glass bottles. These solutions were used to construct calibration curves after appropriate dilution.

### 2.4. Materials

Purple and green AR samples were obtained from a commercial asparagus farm in the South Island of New Zealand (Palmerston, New Zealand). The plants were 15 years old. Yellow, green, purple, and white AR samples, grown for 8–10 years, were obtained from Heze City (Shandong Province, China). The samples were cleaned and rinsed with distilled water several times before freezing at −20 °C. The frozen roots were then freeze-dried using a freeze dryer (ALPHA1–2, Martin Christ Gefriertrocknungsanlagen Co., Ltd., Osterode, Germany), divided into three groups for each variety, pulverized using a micro-plant grinding machine (FZ102, Tianjin Shi Taisite Equipment Co., Ltd., Tianjin, China), and sieved to produce three batches (*n* = 3) of finely ground (<420 nm) samples that were used in further analyses.

### 2.5. UPLC-MS/MS Instrumentation

Ultra-performance liquid chromatography (UPLC)-MS/MS analyses were carried out using an UltiMate 3000 Performance LC^TM^ system (Thermo Fisher Scientific, Hemel Hempstead, UK) linked simultaneously to both a Waters 2996 photodiode array (PDA) detector (Thermo Fisher, UK) and a Micromass Quattro micro^TM^ API benchtop triple-quadrupole mass spectrometer (Thermo Fisher MS Technologies, Manchester, UK) equipped with a Z-spray electrospray ionization (ESI) source operating in positive ion mode. Xcalibur^TM^ software (version 2.2.7, Thermo Fisher Scientific, Hemel Hempstead, UK) was used for data acquisition, data processing, and instrument control. Screwcap bottles (1.5 mL) and hydrophobic filter membranes (0.22 μm, 26 mm) were purchased from Agilent (Palo Alto, CA, USA).

#### 2.5.1. Liquid Chromatography (LC) Separation

LC separation was performed via ion exchange chromatography on a Hypersil GOLD aQ column (250.0 mm × 4.6 mm, particle size 5 µm; Thermo Fisher Scientific, Hemel Hempstead, UK). All runs were performed under gradient elution conditions at a flow rate of 0.25 mL/min using acetonitrile and 0.1% formic acid in methanol-H_2_O as the mobile phase (Table 1). The temperature of the column was set at 30 °C. The volume of injection was 5 µL. The diode array detector (DAD) detected analytes at 278 nm with a reference wavelength of 580 nm (both at 4-nm bandwidth), with full spectral scanning from 280 to 425 nm and 0.5-nm resolution using a semi-micro flow cell.

#### 2.5.2. Mass Spectrometry (MS)

ESI mass spectrometry was performed in positive ion mode (ESI^+^) with multiple reaction monitoring (MRM). The electrospray capillary voltage, extraction voltage, and six polar voltages were 3.5 kV, 30 V, and 0.35 V, respectively. The ion source and desolvation temperatures were set at 110 °C and 300 °C, respectively. The flow of the nebulizer and desolvation gases nitrogen and hydrogen were set at 60 and 50 L/h, respectively. The total post-injection equilibration time was determined to be 25 min, including 20 min for injection and 5 min at the end of the gradient. Table 2 shows the tandem mass spectrometric parameters for caffeic acid, quercetin, apigenin, ferulic acid, baicalein, and kaempferol. Figure 1 shows the MS/MS chromatogram of a mixed surrogate standard (caffeic acid, quercetin, apigenin, ferulic acid, baicalein, and kaempferol).

### 2.6. Sample Preparation and Solid-Phase Extraction (SPE) Procedures

#### 2.6.1. Sample Preparation

Six samples (2 g each) were extracted using a UAE system under optimized conditions (solid-to-liquid ratio of 1:40 with 80 mL of 1.5% formic acid in ethanol–H_2_O (50:50, *v/v*) extracted by UAE at 60 °C with 550 W for 80 min). All the extracts were filtered using a 0.22-μm hydrophobic filter membrane before injection. All samples were analyzed in triplicate.

#### 2.6.2. SPE Preparation

SPE cartridges were prepared by placing four different adsorbents, i.e., a silica-based C-18 solid-phase extraction cartridge (500 mg, 6 mL), an Agilent Strata-C18 Bond Elut cartridge (500 mg, 6 mL), an Oasis HLB column (500 mg, 6 mL), and a Supra-Clean (SI-S) cartridge (500 mg, 6 mL), inside 6-mL polypropylene tubes between two polyethylene frits. Oasis HLB columns were connected to the Agilent Strata-C18 Bond Elut cartridge, Supra-Clean (SI-S) cartridge and silica-based C-18 cartridge by the adapters, and 500 mg of Oasis HLB was placed at the top. Then, 5 mL of methanol and 5 mL of Milli-Q water with a Supelco Visiprep vacuum manifold (Bellefonte, PA, USA) were used to treat the HLB cartridges, and then, both cartridges were dried for 3 min before application of the sample. The elution process was carried out using 5 mL of 0.1% formic acid in methanol-water (70:30, *v/v*) at a flow rate of 3 drops/second (drops/s) under vacuum. Then, 3 mL of the eluted material was dried under nitrogen gas (N_2_) at 30 °C and re-dissolved in 500 μL of 0.1% formic acid in methanol-water (70:30, *v/v*). Then, the solution was filtered through a 0.2-μm nylon filter and transferred to a 1-mL sample vial for UPLC-MS/MS analysis.

### 2.7. Method Validation

The method performance was assessed in terms of the linear range, limit of detection (LOD), limit of quantification (LOQ), stability, repeatability, precision (%RSD), and recovery [13] based on the European Santé et Consommateurs (Directorate General Health and Consumers; European Commission; Brussels, Belgium, commonly known as SANCO) guideline 12571/2013 and Commission Regulation (EC) no. 401/2006 [14].

An MS/MS transition was used to determine the interference peak at the retention time for each compound. The LOQ was dependent on the signal-to-noise ratio, which was investigated in AR matrices spiked with 30, 150, 54, 70, 43.6, and 22.5 ng/mL of caffeic acid, quercetin, apigenin, ferulic acid, baicalein, and kaempferol, respectively. Six calibration levels were constructed for the various compounds (22.5–500 ng/mL for kaempferol, 43.6–500 ng/mL for baicalein, 50–500 ng/mL for ferulic acid, 30–500 ng/mL for apigenin, 150–500 ng/mL for quercetin and 70–500 ng/mL for caffeic acid). The analytical assay was validated by spiking AR extracts at three different concentrations. The concentrations were 140, 280, and 500 ng/kg for caffeic acid; 30, 60, and 120 ng/kg for quercetin; 60, 120, and 240 ng/kg for apigenin; 108, 200, and 400 ng/kg for ferulic acid; 87.6, 175.2, and 350.4 ng/kg for baicalein; and 45, 90, and 180 ng/kg for kaempferol. These extracts were added to the AR samples, which was repeated six times. This method for recovery and RSD determination was performed according to international standards and regulations [15].

#### 2.7.1. Residual Amount Calculation

The caffeic acid, quercetin, apigenin, ferulic acid, baicalein, and kaempferol levels were determined by an external standard method (ESM). The formula used was as follows: *R* = (*A_i_* × *C_s_* × *V_i_*)/(*A_s_* × *W*), where *R* is the extract of the AR sample (mg/kg), *A_i_* is the peak area of the tested material, *A_s_* is the peak area of the standard, *C_s_* is the concentration of the standard (μg/mL), *V_i_* is the constant volume of the AR sample (mL), and *W* is the AR sample weight (g) [13].

The standard curve calibration and single-point correction method were included in the ESM for the correction of matrix effects. The single-point correction assay was performed to calculate the amounts of the tested compounds in the present study. The standard and sample solutions were prepared for analysis under the same LC conditions [13].

#### 2.7.2. Quantitative Analysis

Five concentrations of external standards (12.5, 50, 100, 150, and 250 µg/mL) were used for quantification using least-squares linear regression. Table 3 shows the calibration curves of the tested compounds. Three detection signals were used, namely, the MS-SIM (mass spectrometry-selective ion monitoring) signal, UV-vis response, and MS-extracted ion chromatogram (EIC), where a specific value of *m/z* depending on the [M + H]^+^ of the analyte determined the EIC signal. The response factors (RFs) and regression coefficient (*R*^2^) were used to assess linearity.
$$RF=\frac{DR}{C}$$
where *DR* is defined as the detector response in area counts, and *C* is the concentration of the injected analyte. The best linear calibration curve produced by the detection signal for individual analyses was subsequently selected for quantitative analysis.

### 2.8. Cell Cultures of the HepG2 and L929 Cell Lines

The hepatocellular carcinoma cell line (HepG2) and fibroblast cell line (L929) were provided by the Anatomy Department, University of Otago, Dunedin, New Zealand. Both cell lines were grown in Dulbecco’s modified Eagle’s medium (DMEM; Gibco, Grand Island, NY, USA) supplemented with 10% heat-inactivated fetal bovine serum (FBS; Biological Industries, Cromwell, CT, USA) and 1% antibiotic-antimycotic solution (10,000 U/10 mL penicillin, 10 µg/10 mL streptomycin, 25 µg/mL amphotericin B; Thermo Fisher Scientific, Waltham, MA, USA) in 25-cm^2^ vented cell culture flasks (JET Bio-Filtration Products; Guangzhou, China). Aliquots were placed in 50-mL Falcon tubes (Corning Life Sciences, Corning, NY, USA) and stored at −4 °C. All procedures were conducted under sterile conditions using sterile instruments in a laminar flow hood (Thermo Fisher Science, Waltham, MA, USA). Both cell lines were cultured in a humidified incubator (Thermo Fisher Science, Waltham, MA, USA) at 37 °C with 5% CO_2_. Growth medium was replaced every three days. Once they reached the desired 70–80% confluency, the cells were prepared for sub-culturing under sterile conditions. After the third passage, the cells were used in the experiments [16].

#### 2.8.1. Induction of Oxidative Stress

After both the HepG2 and L929 cells reached the required confluence, the culture medium was substituted with fresh medium containing 0.1% dimethyl sulfoxide (DMSO); 1, 0.5, 0.25, 0.125, 0.0625, or 0.03125 mg/mL different AR cultivars; 10% FBS; and 0.1% (*v/v*) antibiotic-antimycotic solution. HepG2 and L929 cells were cultured for 24 h, and then, the culture medium was removed. Both the HepG2 and L929 cell lines were washed with fresh medium without FBS and exposed to 500 mM H_2_O_2_ for 1 h. Negative control (untreated cells) and positive control (cells + H_2_O_2_) experiments were performed in parallel with cell treatment. Cell viability (MTS) and lactate dehydrogenase (LDH) leakage assays were performed in triplicate.

#### 2.8.2. Determination of Biomarkers of General Cellular Health and Integrity

##### MTS Cell Proliferation Assay

A novel tetrazolium compound [3-(4,5-dimethyl-2-yl)-5-(3-carboxymethoxyphenyl)-2-(4-sulfophenyl)-2*H*-tetrazolium, inner salt; MTS] was used to indicate cell proliferation [16]. Two thousand HepG2 and L929 cells were seeded in each well of 96 plates and incubated at 37 °C for 72 h. After removal of the cell culture medium, 20 μL of MTS reagent was added to each well and incubated for 2 h at 37 °C in a 5% CO2 atmosphere. Cell viability was measured at 570 nm using a micro-plate reader (Perkin Elmer, San Jose, CA, USA). Cell proliferation was expressed in terms of cell number.

##### Lactate Dehydrogenase (LDH) Leakage

LDH leakage was used to evaluate the cell membrane integrity of the HepG2 and L929 cell lines [16]. After treatment, 100 μL of culture medium was added to 100 μL of reaction mixture (Cytotoxicity Detection Kit (LDH); Roche Diagnostics, Mannheim, Germany) and incubated at 25 °C for 30 min. Then, the absorbance of the mixture was measured at 490 nm using a micro-plate reader. LDH activity can be expressed as follows:$$\%\;\mathrm{Cytotoxicity}\;=\;\frac{Compound-treated\;LDH\;activity-Spontaneous\;LDH\;activity}{Maximum\;LDH\;activity-Spontaneous\;LDH\;activity}\times 100$$

### 2.9. Statistical Analysis

All experiments were performed in triplicate, and the results are reported as the mean ± SEM. Variance analyses were performed using the Minitab 16 software package (Minitab Pty Ltd, Sydney NSW 2000, Australia), and *p* < 0.05 was deemed significant.

## 3. Results and Discussion

### 3.1. Determination and Identification of Six Characteristic Compounds in AR

#### 3.1.1. Optimization of Separation of the Six Standard Compounds

Appendix A depict the phytochemical standards (caffeic acid, quercetin, apigenin, ferulic acid, baicalein, and kaempferol) separated at 260, 267, 322, and 360 nm via gradient elution. The order of elution was as follows: the two polar compounds (caffeic acid and ferulic acid) were first eluted at 260 nm, followed by the less polar compounds (apigenin and baicalein, with peaks at 267 nm). In contrast, quercetin and kaempferol were separated at 322 nm and 360 nm, respectively. Caffeic and ferulic acids, eluted at 1.83 and 2.64 min, respectively, have one less flavone hydroxyl group than apigenin and baicalein. Tandem mass spectrometry was optimized under chromatographic running conditions using a mixture of standard solutions of the tested compounds at a concentration rate of 12.5–500 ng/mL. Daughter ion scan acquisition was performed with different collision energies (CE). Precursor ions were selected for optimization of the CE and selection of product ions. The results indicate the absence of a C-5 hydroxyl group, ring, C-11 hydroxyl group, and 2-propenoate ion in phenolic compounds, which are present in caffeic acid and ferulic acid. These compounds exhibit the same intramolecular features, such as a hydrogen bond at the C-3 carbonyl group, while quercetin, apigenin, and baicalein exhibit a C-6 hydroxyl group, and kaempferol exhibits a C-7 hydroxyl group (Appendix A). The presence of a hydroxyl bond appeared to reduce the polarity of the flavones due to chemical interactions of the polar hydroxyl group located at C-5 and a reduction in the polarity of the mobile phase. Therefore, the replacement pattern of the hydroxyl groups affected the elution profile of the flavones.

In this study, screening of multiple AR matrices using this method was performed to demonstrate the potential use of AR matrices in commercial nutraceutical production. Separation of the six compounds was completed within 20 min, with an additional 5 min to re-equilibrate and clean the chromatography column.

#### 3.1.2. Optimization of LC Conditions

Different organic solvents (methanol, ethanol, and acetonitrile (MeCN)) with water as mobile phases were tested for the optimization of chromatographic separation. MeOH provided a narrow and sharp peak, which was better than EtOH (which provided a broad peak); this finding is consistent with a previous study on flavones in *Huangqin* [17]. On the other hand, MeCN provided the sharpest peak; however, suppression of ionization was observed at high concentrations (near 100%). To further optimize the chromatographic separation, formic acid was tested at different concentrations with organic mobile phases. When 0.1% formic acid was used, narrow and sharp peaks were obtained, which was consistent with previous studies conducted on *Huangqin* and *Artemisia annua* L. [17,18]. With increasing percentages of formic acid, the peak width increased and sharpness decreased. Thus, mobile phases containing 0.1% formic acid with both MeOH and MeCN were used for gradient elution via RP-UPLC-MS/MS with a C18 chromatographic column for standard analysis.

With the same gradient of mobile phases, two UPLC chromatographic columns (BEH C18, 1.7 mm, 2.1 mm × 100 mm, and Hypersil GOLD aQ, 250.0 mm × 4.6 mm, 5 µm) were screened for examination of the efficiency of analyte separation. The Hypersil GOLD aQ column was selected as the best column for simultaneous separation and detection of the tested compounds. Appendix A show the narrow and sharp chromatographic peaks obtained with high sensitivity by using the Hypersil GOLD aQ column. A range of flow rates (0.1 to 0.5 mL/min) were screened and 0.25 mg/mL was chosen for separation of the tested compounds.

### 3.2. Optimization of Extraction Procedures with a Clean-Up Step

#### 3.2.1. Clean-Up Procedures

SPE was developed to reduce the matrix effects resulting from the co-eluting residual matrix components. Matrix effects decrease the ionization efficiency of target compounds and reduce the sensitivity of the technique, resulting in erroneous quantitative results [19]. The sample extract was passed through an SPE column and collected to remove co-extracted compounds from the SPE step [19]. The SPE cartridge retained the interfering matrix co-extract while allowing the target analytes to pass through, leading to improvement of separation and purification in a rapid and convenient manner [20]. The recovery of target compounds before and after the SPE procedure was determined using an initial concentration of 100 ng/mL. Appendix A shows the recovery of the six analyte compounds, with values ranging from 34.7% to 64.8%, by a silica C-18 SPE cartridge, as detected using UPLC-MS/MS.

The merits of silica-based C-18, Agilent Strata-C18, and Oasis HLB as suitable adsorbents for the recovery of phenolic compounds, such as caffeic acid, quercetin, rutin, and kaempferol, from plants have been investigated [3,4,20,21]. Supra-Clean (SI-S) columns contain a normal solid-phase medium that can remove matrix components better than an Agilent Strata-C18 Bond Elut cartridge. SI-S and silica C-18 interact with chemicals via hydrogen bonds and remove similar types of compounds, such as structurally similar polar and non-polar compounds. In particular, SI-S cartridges can remove interfering pigments, as evidenced by visual examination. Due to the presence of hydrogen bonds, the different cartridges exhibited different absorption abilities, which affected the recovery of the six compounds (Appendix A). Recovery of caffeic acid using SI-S was the lowest compared to that by the other cartridges. Oasis HLB could be used to recover five compounds from AR (75.4% caffeic acid, 69.2% apigenin, 52.8% ferulic acid, 56.1% baicalein, and 52.8% kaempferol) (Appendix A). Although Oasis HLB achieved satisfactory recovery, the level of interfering pigments was higher than that obtained with SI-S. Generally, Oasis HLB and silica C-18 have been the most widely employed materials for the purification of flavones [20]. However, silica C-18 achieved significantly lower recovery than Oasis HLB for baicalein, apigenin, and kaempferol in the clean-up procedure (*p* < 0.05). The SPE columns with silica C-18 and Oasis HLB exhibited acceptable recovery of the six compounds (85.8% caffeic acid, 90.2% quercetin, 84.0% apigenin, 73.1% ferulic acid, 76.7% baicalein, and 74.5% kaempferol).

Based on these results, the silica C-18 and Oasis HLB adsorbents were found to be suitable for clean-up of the extract solution before UPLC-MS/MS. The clean-up efficiency achieved with a combination of these two SPE columns was consistent with a previous study [19]. A combination of two or three adsorbents has been used as an additional procedure for clean-up [22]. To the best of our knowledge, this study is the first to use a mixture of SPE media to purify phytochemicals from AR matrices in a rapid and simple manner.

#### 3.2.2. Optimization of Extraction Procedures

UAE is more efficient than microwave-assisted extraction (MAE) and conventional extraction methods for extraction of bioactive compounds from plant materials [8]. The extraction efficiency was determined by spiking the AR matrix with the target compound. It was found that 70% acidified aqueous ethanol exhibited a high recovery of target compounds, which is consistent with previous studies on bamboo leaves, *Huangqin* and *Artemisia annua* L. [17,18,23]. Quercetin recovery was achieved at an acceptable level (78.4%), but unfortunately, apigenin and kaempferol recovery was very low (49.1% for apigenin and 43.8% for kaempferol; Figure 3). As reported in the literature [24], increasing the concentration of formic acid within the range 0.1–2% could improve the recovery of bioactive compounds in plant extracts. The use of formic acid improved the recovery of apigenin from 49.1% to 62.7% and that of baicalein from 52.6% to 63.4%.

All target compounds were extracted using 1.5% formic acid in 50% ethanol:H_2_O, achieving high recovery of all compounds (up to 74.7%) except kaempferol (60.4%) (Appendix A). Generally, when a certain amount of an electrolyte is added to a sample solution, the distribution into the organic phase increases and enhances the recovery of compounds due to salting-out effects [24,25,26]. The addition of salting-out reagents, such as NaNO_2_ and AlCl_3_, is an excellent method because of the strong chelating effects of sodium (Na^+^) and aluminium (Al^3+^) ions [27,28]. However, NaNO_2_ or AlCl_3_ could not be used to increase the recoveries of the target compounds in this study because the aim was to produce extracts for functional food. Singaravadivel et al. [5] found that addition of sodium hydroxide (NaOH) was not an efficient method to improve the recovery of caffeic acid and ferulic acid. Therefore, the effects of formic acid on the extraction efficiency were studied by adding 1.5% formic acid to acidified ethanol before centrifugation and comparing the results to those obtained for the extraction solution with 0.1% NaOH. Phytochemical recovery was significantly higher with 1.5% formic acid than with 0.1% NaOH (*p* < 0.05), while the recovery of caffeic acid with 0.1% NaOH was higher than that with 50% ethanol extraction (Appendix A). This difference was observed because apigenin, baicalein, quercetin, and kaempferol are unstable under alkaline conditions. Furthermore, addition of 0.1% NaOH to the extraction solvent resulted in the appearance of a dark yellow color, which was not easy to remove in the clean-up procedure. The use of NaCl, KCl, NH_4_Cl, KH_2_PO_4_, MgSO_4_, and CuSO_4_ in the presence of 0.01–0.1 mol/L NaOH to improve the extraction of caffeic acid from a caffeic acid solution matrix was not successful [24]. However, addition of 0.02 mol/L NaOH alone increased the recovery of caffeic and ferulic acids by 1.8% and 0.74%, respectively [24], which is similar to the findings of this study.

In summary, in this study, 1.5% formic acid was chosen despite a similar level of recovery achieved for caffeic acid by both extraction aids (Appendix A). Optimization of the extraction process using the HLB-SPE cartridge was performed (Section 3.2.1), and a short extraction time was achieved, providing an efficient and effective extraction procedure.

### 3.3. Validation

Table 4 shows the recovery results obtained for the various AR cultivars from triplicate experiments. The recovery of caffeic acid ranged between 81.8% and 88.2%, while the recovery of ferulic acid was between 71.9% and 75.0%. The recovery of quercetin, apigenin, baicalein, and kaempferol was between 68.1% and 101.7%. The RSDs were <10%. There was no blank matrix AR available for evaluation of the recovery rate of endogenous phytochemical compounds. Quercetin, apigenin, baicalein, and kaempferol were selected as surrogate standards because of the structural similarity of these compounds to those in the AR matrix. The recovery of quercetin and apigenin ranged from 87.2 to 95.7% and 68.1 to 101.7%, respectively, in the AR matrix, with RSD < 10% (Table 4). Kaempferol was detected in the third successive extraction. Table 4 shows the repeatability percentages (RSD) for three triplicate extractions with average RSD percentages of 4.9%, 7.5%, 7.1%, 7.7%, 7.7%, and 7.1% for caffeic acid, quercetin, apigenin, ferulic acid, baicalein, and kaempferol, respectively, from AR.

### 3.4. Calibration

The data for the least-squares linear calibration of the six compounds are shown in Table 3. The MS-SIM, MS-EIC, and DAD signals were used to identify these compounds from the AR cultivars. Linearity was determined using *R*^2^ and an RF. Although *R*^2^ is often used to assess regression linearity, *R*^2^ values are more likely to be magnified at low and high calibration concentrations, where there is deviation from linearity. Therefore, RFs, which are not affected by such issues, are suggested to be better measures of linearity [17].

### 3.5. LC-DAD-MS of AR

The six phenolic compounds were identified and quantified from the six AR matrices using both UV/DAD and MS detection. Appendix A shows the UV/DAD and MS-SIM chromatograms of the AR extracts, respectively. The complexity of phenolic compounds in AR varied in both the polar and non-polar areas of the chromatogram as these compounds were are separated from each by gradient elution. The extract chromatogram of AR (Appendix A) indicated a complex polar fraction, which led to increased collection of peaks within the first 5 min. Each compound in the AR matrix was identified using UV-vis and mass spectral characteristics and retention times and comparing the values to those for the standards. For caffeic acid, as a polar compound, the value of λ maximum (λm) was the same as that for ferulic acid, and contributed to the UV absorbance pattern, which demonstrated a lack of chromophores. Caffeic acid and ferulic acid had λm (wavelength) values of 283 and 334 nm, apigenin and baicalein had λm values of 267 and 335 nm, quercetin had a *λ*m value of 267 nm, and kaempferol had a *λ*m value of 335 nm. In addition, the mass spectral pattern (MSP) was identified in the three similar flavone pairs. The MSP of each compound pair differed by only one ion peak: the [M + H]^+^ of caffeic acid (Appendix A). This neutral loss fragmentation pattern was attributed to in-source collision-induced dissociation (CID), which was probably a result of the high fragmentation voltage. In-source CID has been successfully employed for elucidation of various conjugated flavonoid structures, such as glycosylated flavonoids [29]. Neutral loss of C_3_H_4_O_2_ from caffeic acid and ferulic acid in positive and negative modes using single-quadrupole MS was achieved by adjusting the magnitude of the fragmentor voltage. Both apigenin and baicalein had λm values of 322 nm with a [M + H]^+^ peak in the mass spectrum at 271 amu, with the C-6 hydroxyl ion peak at 269 amu, corresponding to the commercial standard with a loss of C_6_H_6_O/C_6_H_6_ from the C-15 position on the A-ring. Similar to the values for quercetin and kaempferol, two λm values, at both 320 and 360 nm, were obtained in the UV-vis spectrum, with [M + H]^+^ peaks at *m*/*z* 153/299 and 153/258, and no fragment ions in the mass spectrum corresponding to apigenin and baicalein.

### 3.6. Concentrations of the Six Compounds in AR

The developed method was employed for detection and quantification of the six compounds (caffeic acid, quercetin, apigenin, ferulic acid, baicalein, and kaempferol) in the AR samples from New Zealand and China. The levels of the six compounds are shown in Figure 2. The results were obtained from triplicate extractions and measurements and exhibited acceptable RSD values (all values were less than 10%). Caffeic acid was the main compound in all six AR varieties, and the highest caffeic acid content was observed in the Chinese yellow AR (5.97 mg/g), which might be attributed to growing region and variety [12,30]. The quercetin content was lower than the caffeic acid content, and quercetin was the second most abundant compound in the AR extracts. The apigenin content was the lowest among the six compounds in Chinese yellow AR. The levels of baicalein and kaempferol were the same in the Chinese yellow AR (*p* > 0.05). The ferulic acid content in all six AR varieties was lower than that reported in the literature [2], which might be attributed to growing conditions, plant age, and other environmental conditions [12,30]. The levels of the other compounds from the AR extracts were significantly lower than those of caffeic acid, which was consistent with the findings reported by Huang et al. [2] who used ultrasonic extraction and a methanol-water (70:30) mixture as a solvent to extract caffeic acid, quercetin, ferulic acid, and kaempferol from AR. The growth conditions and other experimental conditions may have caused some of the observed differences between the two studies. However, the trend of the concentrations, i.e., caffeic acid > quercetin > kaempferol, was consistent in both studies. The concentrations of the other three compounds varied with no obvious trends between the Chinese green and white ARs or New Zealand green and purple ARs (Figure 2).

The phytochemical content in AR matrices is dependent on type, region, climate, age, and other environmental factors [31,32]. Rodríguez et al. [33] investigated white and green *A. officinalis* spears from Alcala del Rio and green, bronze, and purple *A. officinalis* spears from Huetor-Tajar and obtained total flavonoid content (TFC) values of 1.62 mg rutin equivalents (RE)/g of fresh product (FP), 2.25 mg RE/g of FP, 6.40 mg RE/g of FP, 5.26 mg RE/g of FP, and 5.94 mg RE/g of FP, respectively. Kulczyński, et al. [32] studied the total phenolic content in white, green, and pale *A. officinalis* spears from Nowy Tomysl, Poland (Miedzichowo), and obtained total phenolic content (TPC) values of 270–430 mg quercetin equivalents (QE)/100 g of dry weight of extract (DW), 450–730 mg QE/100 g of DW, and 260–420 mg QE/100 g of DW, respectively. These results suggest that different species of AR, and even different parts of AR, contain different phytochemical levels. There were no significant differences in the levels of ferulic acid, baicalein, and kaempferol among the six AR cultivars, except in Chinese yellow AR (*p* < 0.05).

### 3.7. Bio-Protective Capacity of AR Cultivar Extracts

This study is the first to evaluate the in vitro effects of the crude extracts of six AR varieties on HepG2 and L929 cells. Figure 3 and Figure 4 show the potential bio-protective effects of AR extracts, protecting both cell lines against oxidative damage, and a comparison of the extracts obtained from the different cultivars at various concentrations. The Chinese yellow AR extract at high concentrations (1 mg/mL) exhibited significantly stronger protective effects against H_2_O_2_-induced oxidative damage than the Chinese white, purple, or green AR extracts (*p* < 0.05). The effects of low concentrations of AR extracts (≤0.25 mg/mL) were similar to those of the untreated control and the control containing 0.1% DMSO (*p* > 0.05). The likely reason for these observations is the differences in qualitative and quantitative compositions of the phytochemicals in the six AR cultivars (Figure 2). In our previous study, we found that Chinese yellow AR exhibited a significantly higher antioxidant capacity than the other species [30], which probably contributed to the significant protective effect against H_2_O_2_-induced damage in both cell lines. These results suggest that such bio-protective effects on HepG2 and L929 cell proliferation can be attributed primarily to the high levels of caffeic acid and kaempferol extracted under the optimized UAE conditions and secondarily to the higher concentrations of the six phytochemicals in the Chinese yellow AR than in the other cultivars. According to Reference [8], the cell membrane is sensitive to phytochemicals that possess the capacity to eliminate free radical-mediated reactions that cause deleterious oxidative damage in biological cells. In particular, caffeic acid, which was the most abundant phenolic compound in the AR extract from that variety, has the potential to protect DNA from oxidative damage in human lymphocytes [34] and maintain the membrane integrity of rat livers subjected to nickel-induced oxidative damage [35].

Chinese green AR exhibited no enhancement effects in protecting L929 and HepG2 cells against H_2_O_2_-induced damage at low concentrations of 0.25, 0.125, 0.0625, and 0.03125 mg/mL (Figure 3). On the other hand, the New Zealand green AR extracts exhibited significant effects in protecting HepG2 and L929 cell viability and inhibited LDH membrane leakage compared to Chinese green and white AR (*p* < 0.05). The number of viable cells in H_2_O_2_ treated L929 cells by New Zealand green AR at a concentration of 0.25 mg/mL provides protection similar to that of Chinese white AR given at a concentration of 1 mg/mL. Moreover, Chinese green and purple AR significantly protected the L929 cell viability against H_2_O_2_-induced damage (*p* < 0.05). We found that pre-incubation with the highest concentration of Chinese purple AR substantially increased the viability in H_2_O_2_ treated L929 cells. In an effort to understand the effect of protection on the membrane integrity of L929 cells, an LDH membrane leakage assay was used and compared with untreated (negative) counterparts. We found that high concentration of Chinese yellow and New Zealand green AR treatments significantly improved the protective effects in L929 cells. In contrast, low concentrations of Chinese green and white AR extracts did not have a significant effect on cell viability compared with positive control H_2_O_2_ of both cells (*p* > 0.05). This finding indicates that the biological activity of phytochemicals is generally a function of their concentrations, which affect the membrane permeation and interaction with the cellular systems [36]. As the concentration of AR cultivar extracts increased, the L929 and HepG2 cells’ viability and resistance to the oxidative stress induced by H_2_O_2_ were enhanced; these results are in line with the literature [37]. In this context, a concentration of 1.0 mg/mL (crude aqueous extract) of *Osbeckia aspera* leaves was found to exert a more protective effect against induced damage (by bromobenzene and 2,6-diMeNAFQI) to HepG2 cells than lower concentrations (0.25 and 0.5 mg/mL), as reported by Thabrew et al. [2]. This finding was in line with our present study, where Chinese yellow AR (the variety that had the highest phenolic content) provided a better protection of both He G2 and L929 cells against damage induced by H_2_O_2_. Herein, we found that caffeic acid (in Chinese yellow AR) at a concentration of 5.97 mg/g significantly decreased the LDH leakage on H_2_O_2_-induced damage in HepG2 and L929 cells. Interestingly, New Zealand green and purple AR (containing caffeic acid content, i.e., 5.31 and 4.51 mg/g, respectively) exhibited similar levels of LDH release in both cell lines induced by H_2_O_2_ exposure. However, Chinese yellow AR displayed a better bio-protective effect at the highest tested concentration (1 mg/mL), the concentration that completely protected the membrane integrity against oxidative stress in H_2_O_2_-induced HepG2 and L929 cells. This difference could be attributed to the higher photochemical constituents of Chinese yellow AR. Caffeic and ferulic acids protected against oxidative membrane damage owing to their highly conjugated double bond structure and the ability to aggregate in the lipid bilayer located on the cell membrane. Collectively, these findings suggest that phytochemicals in AR have a significant protective influence on cell membrane against H_2_O_2_-induced oxidative damage [38].

## 4. Conclusions

A simple and rapid method for the analysis of phytochemicals in AR cultivars using SPE-UPLC-MS/MS was developed. All six analytes were identified and quantified based on UV-vis data, mass spectral characteristics, and retention times. MS detection exhibited a higher sensitivity for the measurement of low analyte concentrations than DAD detection, and the qualitative establishment of structural features of phytochemical compounds was also performed using MS. In addition, MS facilitated the determination of qualitative fragment patterns of the bioactive compounds, leading to the identification of caffeic acid, quercetin, apigenin, ferulic acid, baicalein, and kaempferol in six different types of AR, suggesting that this method exhibits high sensitivity at low concentrations. Furthermore, the establishment of a biological and physiologically relevant HepG2 and L929 cell model system was important for measurement of the bio-protective capacity of AR by-products by taking into consideration variations in the phytochemicals being obtained from the AR cultivars. The six phytochemicals from AR cultivars that determine the protective capacity towards HepG2 and L929 cells and act against oxidative stress should be isolated and purified in future investigations.

## Figures and Tables

**Figure 1 nutrients-11-00107-f001:**
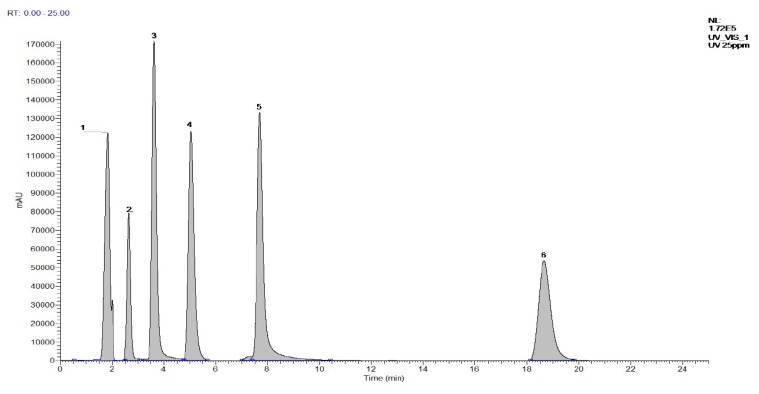
The MRM chromatogram of the six tested compounds in a mixed standard solution (fortified at LOQ level) operated in positive mode. Peaks are in order as follows: 1: Caffeic acid, 2: Quercetin, 3: Apigenin, 4: Ferulic acid, 5: Baicalein, and 6: Kaempferol.

**Figure 2 nutrients-11-00107-f002:**
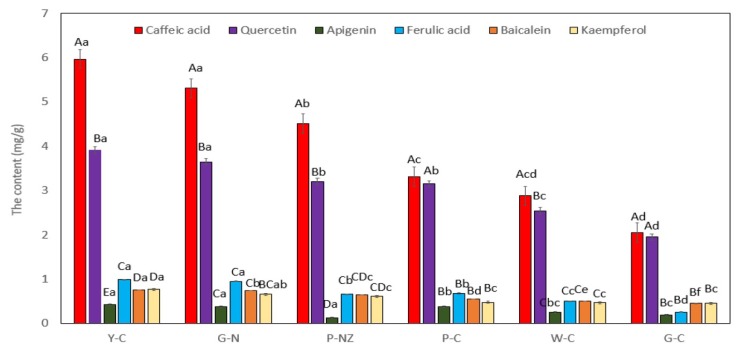
The contents of the six tested compounds in various AR samples. Data presented as mean ± SEM from three replicates (*n* = 3) from each sample. Bars with different letters indicate that they are statistically different (*p* < 0.05). The small letters mean the content of the same compound in different cultivars; whereas, the capital letters indicate the six tested compounds are compared within the same AR. Y-C = yellow Chinese asparagus root; G-N = green New Zealand asparagus root, P-NZ = purple New Zealand asparagus root; P-C = purple Chinese asparagus root; W-C = white Chinese asparagus root; and G-C = green Chinese asparagus root.

**Figure 3 nutrients-11-00107-f003:**
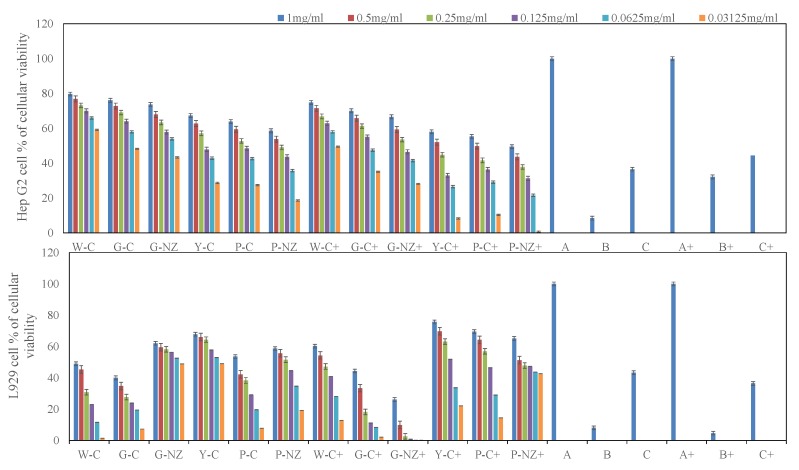
Investigating the bioactive capacity of various concentrations of AR cultivars on the protection of HepG2 and L 929 cells against H_2_O_2_ (hydrogen peroxide) exposure during the MTS assay. Data presented as mean ± SEM from three replicates cell culture experiments (*n* = 3) from each sample. A Positive control: Cell culture medium; C Negative control: 1% SDS (Sodium dodecyl sulfate) medium; B 0.1% DMSO (Dimethyl sulfoxide) control: 0.1% DMSO + cell culture medium. A+ Positive control+: cell culture medium+H_2_O_2_ (500 mM); C+ Negative control+: %SDS +H_2_O_2_ (500 mM); B+ 0.1% DMSO control+: 0.1% DMSO+ cell culture medium+H_2_O_2_ (500 mM).

**Figure 4 nutrients-11-00107-f004:**
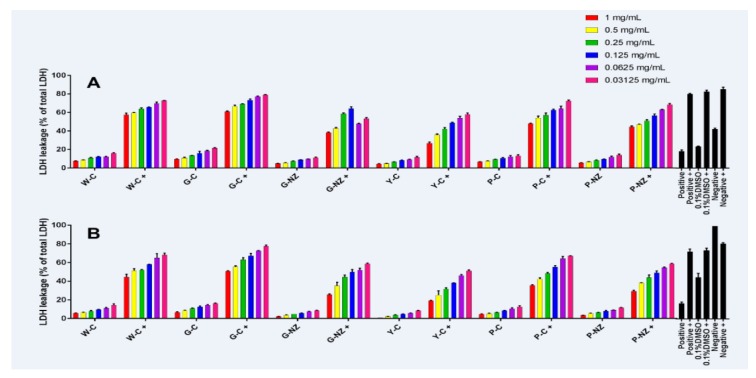
The capacity of different cultivars AR investigated at the different concentrations in protecting HepG2 and L 929 cells from H_2_O_2_ exposure in the LDH assay. Data presented as mean ± SEM from three replicates of the cell culture experiments (*n* = 3) from each sample. Positive control: cell culture medium; Negative control: 1% SDS medium; 0.1% DMSO control: 0.1% DMSO + cell culture medium. Positive control+: cell culture medium + H_2_O_2_ (500 mM); Negative control+: %SDS +H_2_O_2_ (500 mM); 0.1% DMSO control+: 0.1% DMSO+ cell culture medium+H_2_O_2_ (500 mM).

**Table 1 nutrients-11-00107-t001:** Solvent gradient program for UPLC analysis.

Time (min)	Solvent A (%)	Solvent B (%)
0.00	95	5
2.00	85	15
3.00	75	25
4.00	40	60
6.25	55	45
8.25	20	80
20.50	15	95
25.00	95	5

**Table 2 nutrients-11-00107-t002:** The optimized MS parameters for determination of the six components in AR using UPLC-MS/MS.

Components	Parent Ion (*m/z*)	Retention Time (min)	Product Ions for Identification (*m/z*)	Collision Energy (V)	DP * (V)
**Caffeic acid**	180.15	1.83	91/88	20/20	80/80
**Quercetin**	302.24	7.71	153/299	40/40	135/135
**Apigenin**	270.24	5.05	153/119	35/30	80/80
**Ferulic acid**	194.187	2.64	72/123	20/20	80/80
**Baicalein**	271	3.63	123/169	35/35	80/80
**Kaempferol**	287	18.86	153/258	35/30	100/100

* DP: Declustering potential.

**Table 3 nutrients-11-00107-t003:** The regression equation, correlation coefficient, linear ranges, LODs, and LOQs of the six tested components in AR using UPLC-MS/MS.

Analyte	Regression Equation	Correlation Coefficient (*R*^2^)	Linear Range (ng/mL)	LOD (ng/mL)	LOQ (ng/mL)
**Caffeic acid**	*Y* = 5814.9*X* + 94,5763	0.977	70–500	23	70
**Quercetin**	*Y* = 23,142.3*X* + 4 × 10^6^	0.937	150–500	50	150
**Apigenin**	*Y* = 69,288*X* + 694,300	0.996	30–500	10	30
**Ferulic acid**	*Y* = 24,691*X* + 522,209	0.986	50–500	18	54
**Baicalein**	*Y* = 73,699*X* + 1 × 10^6^	0.991	43.6–500	14.4	43.6
**Kaempferol**	*Y* = 75,732*X* + 425,150	0.998	22.5–500	7.5	22.5

*Y*: The peak area ratio of analyte with external standard (ES). *X*: The corresponding concentration for the working standard solutions.

**Table 4 nutrients-11-00107-t004:** Recovery of the six tested compounds in AR.

Standards	Original Quantity (mg/g)	Addition Quantity (μg/g)	Amount Found (mg/g)	Recovery (%)	RSD (%)
**Caffeic acid**	2.02	140	2.16	81.80	4.15
		280	2.30	87.50	3.93
		500	2.49	88.21	6.58
**Ferulic acid**	0.24	108	0.34	75.00	7.37
		200	0.40	71.90	7.68
		400	0.54	72.00	8.09
**Quercetin**	3.05	30	3.10	87.20	6.88
		60	3.12	88.30	7.44
		120	3.18	95.17	8.17
**Apigenin**	0.23	60	0.25	101.70	5.52
		120	0.27	68.10	9.50
		240	0.36	72.10	6.34
**Baicalein**	0.45	87.6	0.52	75.20	5.83
		175.2	0.58	74.70	7.44
		350.4	0.70	80.34	9.85
**Kaempferol**	0.43	45	0.48	76.70	7.46
		90	0.52	73.20	6.37
		180	0.58	73.50	7.52

RSD: Relative standard deviation.

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
