# Peer review of "Identification of Six Phytochemical Compounds from Asparagus officinalis L. Root Cultivars from New Zealand and China Using UAE-SPE-UPLC-MS/MS: Effects of Extracts on H2O2-Induced Oxidative Stress"

_nutrients, 2019, doi:10.3390/nu11010107_

Reviewer 1 Report

In this study, Zhang and colleagues aimed to develop a simple, rapid, specific, and sensitive method for simultaneous identification and quantification of six bioactive compounds (caffeic acid, quercetin, apigenin, ferulic acid, baicalein and kaempferol) present in Asparagus officinalis root, using UAE‐SPE‐UPLC‐MS/MS). Despite the overall interest, quality, and pertinence of this matter, several aspects should be considered:

- title should be shortened, is too long

- l. 34: allowed > allowing

- l. 40: ";" > "("

- l. 49: nutritional value and richness

- l. 68-69: the phrase should be removed from introduction and put into materials and methods section

- l. 84-87: this paragraph should be included in the previous

- l. 88: instruments > methods

- l. 100: which methanol percentage was used?

- table 2: CE - include full legend in the table

- l. 195-197: revise tais sentence

- tables belonging to materials and methods section should be included as supplementary material. in addition, article structure should follows instructions for authors (results and discussion should be included prior materials and methods section)

- figures quality should be improved

- table 4: RSD - include full legend in the table

- figure 5: queritin > quercetin

- figures 6 and 7: please include legend of abbreviations

- the manuscript should be completely revised in order to correct some grammar and spelling typos

Author Response

Comments and Suggestions for Authors

In this study, Zhang and colleagues aimed to develop a simple, rapid, specific, and sensitive method for simultaneous identification and quantification of six bioactive compounds (caffeic acid, quercetin, apigenin, ferulic acid, baicalein and kaempferol) present in Asparagus officinalis root, using UAE‐SPE‐UPLC‐MS/MS). Despite the overall interest, quality, and pertinence of this matter, several aspects should be considered:

- Title should be shortened, is too long

Response: This has been modified to “ Identification of six phytochemical compounds from New Zealand and Chinese Asparagus officinalis L roots cultivars using UAE-SPE-UPLC-MS/MS: Effects of extracts on H2O2-induced oxidative stress”.

- l. 34: allowed > allowing

Response: This has been amended as following “mass spectral pattern of the detected peaks matched with those of commercial standards allowing the characterization of the target compounds” .

- l. 40: ";" > "("

Response: This has been amended as suggested “The content of the target compounds was significantly different (P < 0.05), among the 6 cultivars. Chinese yellow AR had the highest levels of bioactive compounds: 6.0, 3.9, 0.4, 1.0, 0.86, and 0.8 mg/g.

- l. 49: nutritional value and richness

Response: This has been corrected as suggested

Asparagus officinalis L. (green asparagus) is consumed worldwide due to its nutritional value and richness bioactive phenolic compounds.

- l. 68-69: the phrase should be removed from introduction and put into materials and methods section

Response: Amended as suggested “Hepatocellular carcinoma (Hep G2) and mouse fibroblasts (L929) cell lines are used as cell cultures for investigating bio-protective activity in this study” to materials and methods section.

- l. 84-87: this paragraph should be included in the previous

Response: This has been modified to“To achieve this objective, we developed and validated a rapid, simple, specific and sensitive method to quantify six bioactive compounds (caffeic acid, quercetin, apigenin, ferulic acid, baicalein, and kaempferol) in AR using ultrasound-assisted solid-phase extraction coupled to ultra-performance liquid chromatography-tandem mass spectrometry (UAE-SPE-UPLC-MS/MS). Furthermore, the efficacy of the extracts against oxidative stress was examined using an in vitro system.

- l. 88: instruments > methods

Response: Revised

2. Materials and Methods

- l. 100: which methanol percentage was used?

Response: We did use absolute methanol and this has been revised as following “Stock solutions of caffeic acid, quercetin, apigenin, ferulic acid, baicalein and kaempferol at 1000 ng/mL were prepared in absolute methanol.”.

- table 2: CE - include full legend in the table

Response: This has been revised as suggested.

- l. 195-197: Revise tais sentence

Response: This has been revised

- tables belonging to materials and methods section should be included as supplementary material. in addition, article structure should follows instructions for authors (results and discussion should be included prior materials and methods section)

Response: Thanks a bunch for your suggestion. We have moved some of the tables as Suppl. Materials. We have checked the format of the journal and assured that every aspect of the MS was in accordance to GFA.

- figures quality should be improved

Response: Thank you, we have improved the quality of the Figs with high resolution.

- table 4: RSD - include full legend in the table

Response: This has been done

- figure 5: queritin > quercetin

Response: This has been revised, thanks for the good catch.

- figures 6 and 7: please include legend of abbreviations

Response: This has been done

- the manuscript should be completely revised in order to correct some grammar and spelling typos

Response: This has been done throughout the MS

Reviewer 2 Report

Dear authors

The manuscript by Zhang et al. (article ID nutrients-394036), describes the optimized and validate method for simultaneous identification and quantification of the major six bioactive compounds of Asparagus officinalis L. roots from China and New Zealand.

The subject is interesting, it falls within the scope of the journal and it has some novelty. However, the manuscript presents several weaknesses (see comments below and attached file) and is a very preliminary version of a potential manuscript for publication. Please see the points assigned below and in the attached file (not sorted) that could be improved if the authors want kindly address them.

The authors don’t identify two compounds present in all the analyzed extract (15.98 min and 20.60 min) and one of them is the most abundant compound in the extracts.

The structure of the manuscript is not acceptable. I found the following organization:

1. Introduction

2. Materials and Instruments

3. Method validations

4. Cell culture of Hep G2 and L929 cell lines

5. Quantitative analysis

3. Results and discussion

4. Conclusion

 The authors should prepare the manuscript in a much more careful and professional manner. The manuscript show a very confuse work, with a very excessive number of figures, with the most relevant data dispersed among many data not relevant and that they were  attempts to reach the most importante data.

Authors do not need to describe failed attempts. They should describe in detail the optimized method, the identification and quantification of the compounds and the validation of the method.

In the evaluation of bio-protective effect of the extract, the authors should a) present the result as % of cellular viability vs. concentration for each extract instead the cell number vs. concentration; b) show what happens in the tests in which they provoke oxidative stress in the cells, that is the cellular viability of the Hep G2 and L 929 cells after H2O2 exposure only.

It is very difficult to separate the experimental section from the results and discussion. In this last section, I found again, for exemple, the extraction experimental conditions (“…2.0 g of AR extract was mixed with 80 mL of 1.5% formic acid: EtOH: H2O (v/v) for 80 min by UAE at 60with a 550 W. A 5 mL aliquot of the extraction solution was passed through the HLB SPE cartridge…”

 The manuscript text, from the summary to the conclusions needs to be improved. I found phrases whose information seems to be out of the context of the manuscript, overly long phrases that are lost meaning, phrases that refer to the lack of studies on activity and composition of A. officinalis root extract when they exist, non-sequential references, etc. (please see comments and suggestions in attached document).

The authors should be much more rigorous in describing the experimental methods, especially the chromatographic conditions (2.5.1 section). In my opinion, the chromatographic method is not properly optimized or the description presented is incomplete and / or incorrect.

They can not, for example, write that they use an elution under isocratic conditions, when they present a table indicating the variation of the eluent composition over the time!

The last compound elutes at 18.06 min and the last change in eluent composition is at 25 minutes. Why such a long elution program? The authors do not indicate for how long eluates this last eluent. This last change does not correspond to the equilibration conditions of the column in the initial eluent. When does this equilibration occur?

Figure 1 (a) to (f) should be presented as suplemmentary material. Figure 5 is relevant but several of the remaining may be removed or presented as supplementary material.

The solvent used to the compounds extraction from AR is not complete. The authors indicate “…80 mL of 1.5% formic: ethanol: H2O (v/v) for 80 min…). The formic acid (please complete the name) is 1.5 % but, what is the ratio used for etanol and water? This sentence is written several times in the texto (experimental section and discussion of the results).

The authors should try to be consistent in using abbreviations. In the phrase "... been reported for A. racemus, A. cochinchines, Asparagus pubescens, and A. africanus only ...": Why is one of the Asparagus species not abbreviated?

Why do they, throughout the introduction, indiscriminately use Asparagus officinalis root, and the abbreviation AR? The meaning of the abbreviation must be indicated the first time it is used, and then only the abbreviation is used.

The authors write the following sentence “…The overarching objective of this study is to provide useful information on the composition of flavonoids in AR cultivars…” Why do they refer only to the flavonoid profile if the work described in this manuscript identifies flavonoids and hydroxycinnamic acids?

The references should be numbered consecutively in the order in which they are cited in the text. In the introduction, the references jump from ref. [7] to ref [9], while ref. [8] appears only in "3.2.2. Optimization of extraction procedures" section.

The title is too big and contain some detailed information that is not necessary. Please provide a shorter title.

Chemical formula. Please write the stoichiometric coefficient as index. Check the entire manuscript.

The final reference list should be formatted according the Nutrients author instructions (1. Author 1, A.B.; Author 2, C.D. Title of the article. Abbreviated Journal Name Year, Volume, page range, DOI. Available online: URL (accessed on Day Month Year). The references 12 (it is incomplete) and reference 30 are the same paper.

Author Response

Reviewer-2

Comments and Suggestions for Authors

Dear authors

The manuscript by Zhang et al. (article ID nutrients-394036), describes the optimized and validate method for simultaneous identification and quantification of the major six bioactive compounds of Asparagus officinalis L. roots from China and New Zealand.

The subject is interesting, it falls within the scope of the journal and it has some novelty. However, the manuscript presents several weaknesses (see comments below and attached file) and is a very preliminary version of a potential manuscript for publication. Please see the points assigned below and in the attached file (not sorted) that could be improved if the authors want kindly address them.

The authors don’t identify two compounds present in all the analyzed extract (15.98 min and 20.60 min) and one of them is the most abundant compound in the extracts.

The structure of the manuscript is not acceptable. I found the following organization:

1. Introduction

2. Materials and Instruments

3. Method validations

4. Cell culture of Hep G2 and L929 cell lines

5. Quantitative analysis

3. Results and discussion

4. Conclusion

 Response: The main sections have been revised as suggested. Subsection/heading had to be included to help the reader follow the methodology.

The authors should prepare the manuscript in a much more careful and professional manner. The manuscript show a very confuse work, with a very excessive number of figures, with the most relevant data dispersed among many data not relevant and that they were attempts to reach the most important data.

Response: Thanks to the diligent reviewer for raising this query. Indeed, our study comprehends various aspects, including preparation, identification of several compounds from 6 different asparagus varieties as well as biological activity.  That is why there were lots of figures. We could easily divide the MS into two; however, we do prefer to have it in the form of single solid MS rather than diluted one.

Authors do not need to describe failed attempts. They should describe in detail the optimized method, the identification and quantification of the compounds and the validation of the method.

Response: In most cases, these are initial steps that were conducted before reaching to optimal conditions/parameters. Indeed, that is applicable for all analytical methods. Unless otherwise, the method is not innovative. We do believe that this is important part for science communication.

 In the evaluation of bio-protective effect of the extract, the authors should a) present the result as % of cellular viability vs. concentration for each extract instead the cell number vs. concentration; b) show what happens in the tests in which they provoke oxidative stress in the cells, that is the cellular viability of the Hep G2 and L 929 cells after H2O2 exposure only.

Response: Thank you for your comments, This will be done once we get back the files back from a computer company helping us to recover files that have been lost due to computer problems.

 It is very difficult to separate the experimental section from the results and discussion. In this last section, I found again, for exemple, the extraction experimental conditions (“…2.0 g of AR extract was mixed with 80 mL of 1.5% formic acid: EtOH: H2O (v/v) for 80 min by UAE at 60 with a 550 W. A 5 mL aliquot of the extraction solution was passed through the HLB SPE cartridge…”

 Response:. This part has been deleted. 

 The manuscript text, from the summary to the conclusions needs to be improved. I found phrases whose information seems to be out of the context of the manuscript, overly long phrases that are lost meaning, phrases that refer to the lack of studies on activity and composition of A. officinalis root extract when they exist, non-sequential references, etc. (please see comments and suggestions in attached document).

Response: Thank you. This has been amended

The authors should be much more rigorous in describing the experimental methods, especially the chromatographic conditions (2.5.1 section). In my opinion, the chromatographic method is not properly optimized or the description presented is incomplete and / or incorrect. They can not, for example, write that they use an elution under isocratic conditions, when they present a table indicating the variation of the eluent composition over the time! The last compound elutes at 18.06 min and the last change in eluent composition is at 25 minutes. Why such a long elution program? The authors do not indicate for how long eluates this last eluent. This last change does not correspond to the equilibration conditions of the column in the initial eluent. When does this equilibration occur?

Response: Thank you for your comments, in this case with an additional 5 min to re-equilibrate and clean the chromatography column.

 Figure 1 (a) to (f) should be presented as supplementary material. Figure 5 is relevant but several of the remaining may be removed or presented as supplementary material.

Response: I have removed Figure 1 (a) to (f), Figs 2 to 4 to the supplementary materials.

The solvent used to the compounds extraction from AR is not complete. The authors indicate “…80 mL of 1.5% formic: ethanol: H2O (v/v) for 80 min…). The formic acid (please complete the name) is 1.5 % but, what is the ratio used for etanol and water? This sentence is written several times in the tex to (experimental section and discussion of the results).

Response: I have corrected the information as following:

(solid to liquid ratio of 1:40 with 80 mL of 1.5% formic acid in ethanol-H2O (50:50 v/v) for 80 min by UAE at 60 ℃ with 550 W).

The authors should try to be consistent in using abbreviations. In the phrase "... been reported for A. racemus, A. cochinchines, Asparagus pubescens, and A. africanus only ...": Why is one of the Asparagus species not abbreviated?

Response: I have revised all the abbreviation of asparagus species.

the bio-protective capacity of asparagus root extracts have been reported for A. racemosus, A. cochinchines, A. pubescens, and A. africanus [12]

Why do they, throughout the introduction, indiscriminately use Asparagus officinalis root, and the abbreviation AR? The meaning of the abbreviation must be indicated the first time it is used, and then only the abbreviation is used.

Response: I have indicated the abbreviation AR (Asparagus officinalis root) at the first time and also the abbreviation is used only in text.

The authors write the following sentence “…The overarching objective of this study is to provide useful information on the composition of flavonoids in AR cultivars…” Why do they refer only to the flavonoid profile if the work described in this manuscript identifies flavonoids and hydroxycinnamic acids?

Response: I have revised to “The overarching objective of this study is to provide useful information on the composition of bioactive compounds in AR cultivars available”.

The references should be numbered consecutively in the order in which they are cited in the text. In the introduction, the references jump from ref. [7] to ref [9], while ref. [8] appears only in "3.2.2. Optimization of extraction procedures" section.

Response: I have numbered the references consecutively in the right order.

The title is too big and contain some detailed information that is not necessary. Please provide a shorter title.

Response: I have modified to Identification of six phytochemical compounds from New Zealand and Chinese Asparagus officinalis L roots cultivars using UAE-SPE-UPLC-MS/MS: Effects of extracts on H2O2-induced oxidative stress.

 Chemical formula. Please write the stoichiometric coefficient as index. Check the entire manuscript.

Response: I have revised as suggested.

 The final reference list should be formatted according the Nutrients author instructions (1. Author 1, A.B.; Author 2, C.D. Title of the article. Abbreviated Journal Name Year, Volume, page range, DOI. Available online: URL (accessed on Day Month Year). The references 12 (it is incomplete) and reference 30 are the same paper.

Response: I have revised to the relevant references as suggested.

Reviewer 3 Report

In this study, the authors investigated the simultaneous identification and quantification of the major six bioactive compounds, namely (caffeic acid, quercetin, apigenin, ferulic acid, baicalein and kaempferol in Asparagus officinalis root to New Zealand (2 cultivars) and China (Yellow, green, purple and white) using ultrasound assisted extractionsolid phase extractionultraperformance liquid chromatographytandem mass spectrometry (UAESPEUPLCMS/MS). They reported the extracts showed protective effects against oxidative stress in Hep G2 and L929 cell lines.

The topic is within the scope of this journal.

1. Authors need to analyze some more biological activities such as antioxidant, antimicrobial etc.

2. English of the MS needs to be greatly improved. The English of the whole article has to be checked carefully to eliminate linguistic errors.

3. In introduction and discussion part author needs to cite more recent references.

4. Result is enough but discussion is very poor.

Author Response

In this study, the authors investigated the simultaneous identification and quantification of the major six bioactive compounds, namely (caffeic acid, quercetin, apigenin, ferulic acid, baicalein and kaempferol in Asparagus officinalis root to New Zealand (2 cultivars) and China (Yellow, green, purple and white) using ultrasound assisted extraction‐solid phase extraction‐ultra‐performance liquid chromatography‐tandem mass spectrometry (UAE‐SPE‐UPLC‐MS/MS). They reported the extracts showed protective effects against oxidative stress in Hep G2 and L929 cell lines.

The topic is within the scope of this journal.

1. Authors need to analyze some more biological activities such as antioxidant, antimicrobial etc.

Response: Thank you for your comments. We have done the antioxidant and actimicrobial activities in six AR cultivars, which have been submitted and published in the other journal.

 Zhang, H., Birch, J., Yang, H., Xie, C., Kong, L., Dias, G., & Bekhit, A. E. D. (2018). Effect of solvents on polyphenol recovery and antioxidant activity of isolates of Asparagus Officinalis roots from Chinese and New Zealand cultivars. International Journal of Food Science & Technology, 53 (10), 2369-2377

2. English of the MS needs to be greatly improved. The English of the whole article has to be checked carefully to eliminate linguistic errors.

Response: I have revised as you suggested.

3. In introduction and discussion part author needs to cite more recent references.

Response: Thank you for your comments. We could not find more recent references that have been done in this area to our knowledge.

4. Result is enough but discussion is very poor.

 Response: I have revised as you suggested.

Reviewer 4 Report

The article “Identification of six phytochemical compounds from New Zealand and Chinese asparagus officinalis L roots cultivars using ultrasound assisted extraction solid phase extraction coupled with ultra-performance liquid chromatography-tandem mass spectrometry: Effects of extracts on hydrogen peroxide-induced oxidative stress” by H. Zhang and colleagues describes the validation of an UPLC-MS/MS method for the determination of 6 biomolecules in AR. They also investigate their antioxidant effect on HepG2 and L929 cells. The article is well organized and quite well written although several English and editing errors are present. More specifically:

Line 124: “isocratic conditions” is not true, soon after the authors describe the gradient used for the analysis. Isocratic conditions assume that the concentrations of the solvents used as eluent does not change over time.

Line 170: missing closed bracket “)”.

Line 229: untreated cells should be the negative control, while cells treated with H2O2 but without AR should be the positive control. The same applies to section 3.7.

Line 325: Delete “the probably reason is”, probably another error from copy-paste. Did the authors proof-read the paper before submitting it?

Lines 410 and 419: it is Figure 4 (not 3). Similarly, referencing to figures often does not reflect what is in the main text also in the remaining article (e.g. Fig. 4->5, Fig. 5->6, etc).

Line 435: it is Figures 1 and 2. Similarly later in the text. Referencing to the Figures is often wrong and must be checked again.

Line 598: caffeic acid is not hydrophobic; actually, it is quite soluble in H2O, especially if warm H2O, because of the high number of hydrogen bonds that it can form (although the aromatic ring and the conjugated double bond).

Retention times in Figure 1(g) do not agree with those reported in Table 2 (and Figure 2). Similarly, parent ion values in chromatograms in Figure 1(a-f) are slightly different from those reported in Table 2.

Table 5 is useless in the present form (analytes retention times in different varieties) and it does not report what is said in the text (line 560: the content of the different phytochemicals).

How the authors explain the fact that no differences were observed among positive and negative controls in the L929 MTS assay (Figure 6B)?

Revise English, especially in section 3. Missing verbs and weird sentences probably because of copy-paste.

Several editing errors: missing spaces, too many spaces, Celsius degree symbol, etc.

Also revise the bibliography and format it consistently (e.g. missing pages in ref. 12).

Author Response

Reviewer-3

Comments and Suggestions for Authors

The article “Identification of six phytochemical compounds from New Zealand and Chinese asparagus officinalis L roots cultivars using ultrasound assisted extraction solid phase extraction coupled with ultra-performance liquid chromatography-tandem mass spectrometry: Effects of extracts on hydrogen peroxide-induced oxidative stress” by H. Zhang and colleagues describes the validation of an UPLC-MS/MS method for the determination of 6 biomolecules in AR. They also investigate their antioxidant effect on HepG2 and L929 cells. The article is well organized and quite well written although several English and editing errors are present. More specifically:

Line 124: “isocratic conditions” is not true, soon after the authors describe the gradient used for the analysis. Isocratic conditions assume that the concentrations of the solvents used as eluent does not change over time.

Response: I have changed to gradient elution.

Line 170: missing closed bracket “)”.

Response: I have revised to “SPE cartridges were prepared by placing four different adsorbents i.e. Silica-based C-18 solid-phase extraction cartridge (500 mg, 6 mL)” as suggested

Line 229: untreated cells should be the negative control, while cells treated with H2O2 but without AR should be the positive control. The same applies to section 3.7.

Response: Thank you for the comment. We fixed the sentence to reflect that fact. To avoid confusion in the figures we opted to describe the treatments rather referring to them as positive/negative control since we have several controls for in the study as following:

Negative control: cell culture medium;

0.1% DMSO control: 0.1% DMSO + cell culture medium.

Positive control 1: cell culture medium+H2O2 (500 mM);

Positive control 2 for 0.1% DMSO control+: 0.1% DMSO+ cell culture medium+H2O2 (500 mM).

Thank you for the comment. Our understanding that negative control should result in a negative result not expected in normal circumstances (such as the addition of SDS in our case and thus the positive control should be the untreated cells.

 Line 325: Delete “the probably reason is”, probably another error from copy-paste. Did the authors proof-read the paper before submitting it?

Response: I have deleted as suggested.

Lines 410 and 419: it is Figure 4 (not 3). Similarly, referencing to figures often does not reflect what is in the main text also in the remaining article (e.g. Fig. 4->5, Fig. 5->6, etc).

Response: I have corrected all the Figure number referencing to the main text.

Line 435: it is Figures 1 and 2. Similarly later in the text. Referencing to the Figures is often wrong and must be checked again.

Response: I have corrected all the Figure number referencing to the main text.

Line 598: caffeic acid is not hydrophobic; actually, it is quite soluble in H2O, especially if warm H2O, because of the high number of hydrogen bonds that it can form (although the aromatic ring and the conjugated double bond).

Response: I have revised as suggested.

Retention times in Figure 1(g) do not agree with those reported in Table 2 (and Figure 2). Similarly, parent ion values in chromatograms in Figure 1(a-f) are slightly different from those reported in Table 2.

Response: I have putted the right Figs as suggested.

Table 5 is useless in the present form (analytes retention times in different varieties) and it does not report what is said in the text (line 560: the content of the different phytochemicals).

Response: I have deleted table 5 as suggested.

How the authors explain the fact that no differences were observed among positive and negative controls in the L929 MTS assay (Figure 6B)?

Response: Thank you for your comments. These results are now in figure 4. It is clearly there are significant differences between the positive in negative controls in Fig 4B.

Revise English, especially in section 3. Missing verbs and weird sentences probably because of copy-paste.

Response: I have revised as suggested.

Several editing errors: missing spaces, too many spaces, Celsius degree symbol, etc.

Response: I have edited through the main text as suggested

Also revise the bibliography and format it consistently (e.g. missing pages in ref. 12).

Response: I have modified all the cited references and format it consistently.

Round  2

Reviewer 1 Report

All the required aspects were considered by authors.

Author Response

Dear Reviewer

Thank you for your comments. I have done alguage spell check through the whole manuscript.

Reviewer 2 Report

Dear authors

 The manuscript by Zhang et al. (article ID nutrients-394036) was improved and is now a much better manuscript then before. However, the authors should take in to account the points assigned below and already pointed in the first review report.

-          In the evaluation of bio-protective effect of the extract, the authors should present the result as % of cellular viability vs. concentration for each extract instead the cell number vs. concentration.

-          The authors should be rigorous, describing the experimental methods, especially the chromatographic conditions (2.5.1 section, table 1). The authors change the text (line 280: “…with an additional 5 min to re‐equilibrate…”, but they didn’t change the table 1 accordingly. Re-equilibrate means the same eluent used at the first minutes.

-          Chemical formula. Please write the stoichiometric coefficient as index. Check the entire manuscript (see H2O2, for example. It should be H2O2)

-          The final reference list should be formatted according the Nutrients author instructions (1. Author 1, A.B.; Author 2, C.D. Title of the article. Abbreviated Journal Name Year, Volume, page range, DOI. Available online: URL (accessed on Day Month Year).

-          The references 12 and 30 are the same reference. Please delete ref 30 and correct all the following references.

 Author Response

-          In the evaluation of bio-protective effect of the extract, the authors should present the result as % of cellular viability vs. concentration for each extract instead the cell number vs. concentration.

Response: Thank you for your comments, I have modified to the result as % of cellular viability as you suggested.

-          The authors should be rigorous, describing the experimental methods, especially the chromatographic conditions (2.5.1 section, table 1). The authors change the text (line 280: “…with an additional 5 min to re‐equilibrate…”, but they didn’t change the table 1 accordingly. Re-equilibrate means the same eluent used at the first minutes.

Response: Thank you for your comments, I have modified as you suggested.

-          Chemical formula. Please write the stoichiometric coefficient as index. Check the entire manuscript (see H2O2, for example. It should be H2O2)

Response: Thank you for your comments, I have modified as you suggested.

-          The final reference list should be formatted according the Nutrients author instructions (1. Author 1, A.B.; Author 2, C.D. Title of the article. Abbreviated Journal Name Year, Volume, page range, DOI. Available online: URL (accessed on Day Month Year).

Response: Thank you for your comments, I have modified as you suggested.

-          The references 12 and 30 are the same reference. Please delete ref 30 and correct all the following references.

 Response: Thank you for your comments, I have modified as you suggested.

 Reviewer 3 Report

All chromatogram picture need to show in figures, should add the chromatogram picture.

Requested revision carried out by authors. 

Author Response

Dear Reviewer

Thank you for your comments. I have followed the others reviewers' suggestion to remove all the chromatogram pictures as the supplementary figures.

 Reviewer 4 Report

The revised version of the article is significantly improved.

Author Response

Dear Reviewer

Thank you for your comments.